# Global Need for Physical Rehabilitation: Systematic Analysis from the Global Burden of Disease Study 2017

**DOI:** 10.3390/ijerph16060980

**Published:** 2019-03-19

**Authors:** Tiago S. Jesus, Michel D. Landry, Helen Hoenig

**Affiliations:** 1Global Health and Tropical Medicine & WHO Collaborating Center on Health Workforce Policy and Planning, Institute of Hygiene and Tropical Medicine-NOVA University of Lisbon, Rua da Junqueira 100, 1349-008 Lisbon, Portugal; 2Physical Therapy Division, Department of Orthopeadic Surgery, School of Medicine, Duke University, Durham, NC 27705, USA; mike.landry@duke.edu; 3Duke Global Health Institute (DGHI), Duke University, Durham, NC 27710, USA; 4Physical Medicine and Rehabilitation Service, Durham Veterans Administration Medical Center, Durham, NC 27705, USA; helen.hoenig@va.gov; 5Division of Geriatrics, Department of Medicine, Duke University Medical Center, Durham, NC 27710, USA

**Keywords:** rehabilitation, global health, disability, global burden of disease, health services needs and demand

## Abstract

*Background*: To inform global health policies and resources planning, this paper analyzes evolving trends in physical rehabilitation needs, using data on Years Lived with Disability (YLDs) from the Global Burden of Disease Study (GBD) 2017. *Methods*: Secondary analysis of how YLDs from conditions likely benefiting from physical rehabilitation have evolved from 1990 to 2017, for the world and across countries of varying income levels. Linear regression analyses were used. *Results*: A 66.2% growth was found in estimated YLD Counts germane to physical rehabilitation: a significant and linear growth of more than 5.1 billion YLDs per year (99% CI: 4.8–5.4; *r*^2^ = 0.99). Low-income countries more than doubled (111.5% growth) their YLD Counts likely benefiting from physical rehabilitation since 1990. YLD Rates per 100,000 people and the percentage of YLDs likley benefiting from physical rehabilitation also grew significantly over time, across locations (all *p* > 0.05). Finally, only in high-income countries did Age-standardized YLD Rates significantly decrease (*p* < 0.01; *r*^2^ = 0.86). *Conclusions*: Physical rehabilitation needs have been growing significantly in absolute, per-capita and in percentage of total YLDs. This growth was found globally and across countries of varying income level. In absolute terms, growths were higher in lower income countries, wherein rehabilitation is under-resourced, thereby highlighting important unmet needs.

## 1. Introduction

Worldwide rehabilitation needs are growing in tandem with global population growth, population aging, and higher survival rates for people with severe health conditions and disability [1,2,3]. However, to our knowledge, there is no up-to-date, systematic analysis quantifying worldwide epidemiological trends for the whole set of health conditions likely benefiting from physical rehabilitation. In this paper, we analyze how the global needs for physical rehabilitation have evolved from 1990 to 2017, using data from the Global Burden of Disease Study (GBD) 2017.

The GBD study, which has incorporated data since 1990, is the most comprehensive global epidemiological study to date. It combines data from the best available sources (e.g., scientific literature, official statistics from ministries of health, household surveys, vital registries, hospital data, claims data [4,5]), processed with increasingly sophisticated modelling approaches (e.g., a continuously updated Bayesian meta-regression tool) to provide rigorous global, regional, and national estimates of relevant population health. The data sources used to ground the estimates are colossal and have increased for each cycle of the GBD study. Most recently, the GBD 2017 used a total of 15,449 scientific literature sources (6.5% and 47,4% more than the GBD 2106 and GBD 2015 cycles, respectively), 21,100 sources of epidemiological surveillance data (12.3% and 49.9% more than GDB 2016 and GBD 2015, respectively), and data from an additional 148,842,107 hospital admissions globally, bringing the total number of admissions that inform GBD estimation to more than 2.6 billion [5,6].

For the first time in the study’s history, the GBD 2017 reports data on injuries in terms of their nature or consequence (e.g., hip fracture or spinal cord injury), in addition to and apart from their cause (e.g., falls, road injuries, interpersonal violence) [6]. This new type of data is particularly germane to the planning of services and resources in physical rehabilitation.

Others have used GBD study data to examine rehabilitation needs [7,8]. In the more recent example, the World Health Organization (WHO) used data from the GBD 2015 to examine worldwide needs for mental and physical rehabilitation. They found a 17.6% increase from 2005 to 2015 in Years Lived Disability (YLDs) for health conditions associated with severe disability, and that a remarkable 75% of the total world’s YLDs in 2015 came from health conditions germane to rehabilitation [7]. However, the WHO study did not examine physical rehabilitation needs distinct from the rehabilitation of mental health conditions. Moreover, health conditions with mild disability weights were not considered.

In addition to the WHO’s analysis, a number of studies have used GBD data to examine particular conditions likely benefiting from physical rehabilitation, such as injuries (i.e., their causes) [9], musculoskeletal conditions [10,11], neurological disorders [12,13], stroke [14], cardiovascular diseases [15], chronic respiratory diseases [16], cancer [17], or HIV [18]. However, to our knowledge, no study has used current GBD data to examine YLDs and potential needs for rehabilitation for the broad spectrum of conditions potentially benefiting from physical rehabilitation.

Using publicly available data from the GBD 2017, this paper aimed to provide a systematic global analysis of YLDs for the combined set of health conditions likely benefiting from physical rehabilitation, and specifically answer the following study questions:How large are the estimated world’s YLDs (i.e., YLD counts, YLDs rates per 100,000 peole, and age-standardized YLD rates) likely benefiting from physical rehabilitation in 2017?What is the percentage of the world’s YLDs likely benefiting from physical rehabilitation relative to total YLDs?Did any of these estimates change significantly over time (i.e., since 1990)? If so, by how much? and, finally,Did any of those trends differ across countries of varying income levels?

The answer to these questions would provide valuable benchmarks to inform global and local health and rehabilitation policies, research, and advocacy. Moreover, such findings could provide direction to funding priorities, as well as timely information for health system planning and strengthening activities, including for determining needs for physical rehabilitation services, as a means to allocate resources accordingly.

## 2. Methods

### 2.1. Study Design

Systematic, secondary analysis of public-domain data from the GBD 2017, extracted from the freely-available web platform: the Global Health Data Exchange tool (http://ghdx.healthdata.org/gbd-results-tool).

### 2.2. Data Selection

From the list of YLD “causes” (i.e., underlying health conditions), “impairments” (i.e., consequences of injury or diseases that have more than one diagnostic cause) and “injuries” (i.e., the nature of the injury), the authors selected those deemed likely to benefit from physical rehabilitation, as described below. Given that YLDs from “causes” are collective exhaustive, to avoid double counting, we do not cumulatively select YLDs from Injuries as a “cause”, but only YLDs from the nature of the injury. In addition, YLDs from “impairments” were limited to those not coming from “causes” we had already selected.

We found no global standard to inform which health conditions potentially benefit from physical rehabilitation; thus, in determining which conditions to focus on as potentially benefitting from physical rehabilitation, we relied on the following: previous works using GBD data for rehabilitation [7,8,19]; papers analyzing publication trends in physical rehabilitation and physical therapy across conditions [20,21,22]; and finally on the below working definition [23,24,25,26,27,28].

Physical rehabilitation uses health-based approaches to optimize physical function and participation in persons with physical impairments (e.g., mobility) or symptoms (e.g., low back pain) that are amenable to recovery, prevention, or management from physical rehabilitation services, facilities or providers (e.g., physical or occupational therapists). From this definition, we excluded rehabilitation of people with oral, mental health, substance abuse disorders, intellectual or sensory impairments per se, although partial overlap often occurs across different forms of rehabilitative interventions for those conditions. Table 1 provides the full list of the included conditions.

Of note, within the application of the working definition and exclusion criteria, ‘dubious’ selection decisions existed, e.g., health conditions for which it was not certain whether physical rehabilitation would be applicable or likely to provide benefits. For example, in Autism Spectrum Disorder, psychosocial disfunction and rehabilitative approaches are often dominant, but sensory-processing and motor impairments are prevalent and might benefit from physical rehabilitation as well [29,30,31,32,33,34,35]. In this context, we included this condition, although this choice may lead to an over-estimation of physical rehabilitation needs. By contrast, we excluded some other health conditions, e.g., those for which the benefit of rehabilitation has not been fully established yet. To reduce the bias towards over- vs. under-estimation resulting from ‘dubious’ selection decisions, a priori evidence-based reasoning was established for the conjunct of ‘dubious’ selection decisions (see Appendix B [29,30,31,32,33,34,35,36,37,38,39,40,41,42,43,44,45,46,47,48,49,50,51,52,53,54,55,56,57,58,59,60,61,62,63,64,65,66,67,68,69,70,71,72,73,74,75,76,77,78,79]).

Indeed, toward informing these decisions, we searched PubMed primarily for aggregated evidence, preferably systematic reviews, on the effect of physical rehabilitation approaches (e.g., physical therapy, cardiac rehabilitation, pulmonary rehabilitation) on each type of condition we identified as potentially “dubious” for inclusion. We further gave priority to locating and selecting systematic reviews from Cochrane Database of Systematic Reviews, whenever available (i.e., adding “AND “Cochrane Database Syst Rev” [Journal]” to the search approach). As an example, we found that the quantity, quality and strength of the evidence base for cardiac rehabilitation varies across various cardiac conditions [36,37,38,39,40,41,42], but strong evidence that cardiac rehabilitation approaches are overall cost-effective [44]. Thus, we included YLDs from all cardiovascular conditions as likely benefiting from physical rehabilitation. As another example, we did not include any of the maternal, urological or gynecological conditions, even though there is a body of literature pointing to the fact that physical therapy can be useful for addressing incontinence and pelvic pain [70,71]. However, we found that underlying evidence based on the use of physical therapy for the chronic pelvic pain and incontinence is still considered insufficient [72,75], and the vast majority of urological and gynecological conditions are not typically treated with physical rehabilitation. In addition, data weren’t available in the GBD to identify particular urological/gynecological conditions potentially benefiting from rehabilitation, i.e., there are no GBD “impairments” for consequences of urological/gynecological conditions such as incontinence or pelvic pain. Thus, we excluded the maternal, urological or gynecological diseases altogether, even though some might generate YLDs eventually benefiting from physical rehabilitation.

Finally, among the set of conditions in Table 1, we have selected a few (i.e., a subgroup) for a separate analysis. This subgroup consisted of a narrow, highly conservative set of health conditions historically associated with physical rehabilitation need, i.e., musculoskeletal and key neurological diseases and injuries, including stroke, which frequently cause mobility impairments. Table 2 details the exact subgroup of conditions selected for that separate analysis.

### 2.3. Measures and Data Extraction:

Among the measures available in the webtool, we only extract data for *YLDs*, i.e., the aggregative measure of the GBD study that focuses exclusively in non-fatal impacts of health conditions. YLDs are the years lived with any short-term or long-term health loss weighted for severity by the disability weights. Concretely, YLDs are computed by combining the estimated prevalence of a health condition with the estimated number of years people typically live with those sequelae, up to the population life expectancy threshold. Importantly, the prevalence of sequelae from each condition is classified by severity levels (e.g., mild, moderate, severe), each one having a respective disability weight [5,6,80,81]. For stroke, for example, disability weights vary from 0.019 for mild consequences to 0.588 for severe consequences plus cognition problems, as determined by population preferences in large discrete choice experiments [81]. Details on how YLDs and disability weights are determined, and the current disability weights for all conditions, are provided elsewhere [5,6,81].

For our metrics, we extracted YLD data for prevalent *number* (YLD counts), *rate* (per 100,000 people), and *percentage* (percentage of YLDs from the selected conditions relative to the total amount of YLDs). For age, we extracted YLDs both for *all ages* and *age-standardized* rates. With those specifications, we obtained YLDs likely benefiting from physical rehabilitation under the following forms, used as measures for the analysis:YLD Counts: Nominal amount of YLDs;YLD Rates: YLDs per 100,000 people, i.e., adjusted for population size only;Age-standardized YLD Rates: YLD values adjusted for both population size and ageing;Percentage of YLDs Benefiting from Physical Rehabilitation: The proportion of YLDs from conditions likely benefiting from physical rehabilitation among total YLDs, provided in percent values.

With regards to location, YLDs were extracted at the global level (i.e., for the *world*) and for *high*, *upper middle*, *lower middle*, and *low-income* countries, according to the World Bank Income Levels. We also extracted YLDs for all the years from 1990 to 2017, and for each selected health condition (Table 1).

Using the measures and specifications above, data extraction occurred in early December 2018. All of the selected data were imported from the webtool to Excel spreadsheets for data storage, management, and analysis.

### 2.4. Data Analysis

To determine the overall physical rehabilitation needs, we combined (i.e., summed) the YLDs data for all the selected health conditions—within each of the “years”, “locations”, and “metrics”. Percent changes from 1990 to 2017 were also computed for the combined values.

For inferential statistics, first we plotted the entire time series of the combined values for each location, within each YLD metric. Using visualization and *r*^2^ values of pilot regression models, then we determined which regression model type (i.e., linear, exponential, or logarithmic) best fit the plotted data, and recorded the respective *r*^2^ values. Given differences in *r*^2^ values <0.02 between the models, we retained the linear regression option. Linear regression analyses, using the analysis of variance (ANOVA)—which account for the data on every year between 1990 and 2017, then were used to test for statistical significance, notably of a non-zero yearly change, considering the full time-period between 1990 and 2017, inclusively. We used the linear regression analytical approach even for the data that fitted an exponential or logarithmic-type of regression. In those cases, we tested the models using either the actual YLD values or a ‘log-transformed’ version of these. As there was no difference in the statistical significance for any case, we retained linear regression analyses using the actual YLD values. This facilitated the interpretation of the results since the coefficient and their confidence intervals refer to the estimated annual change in YLD values, i.e., the slope of the linear regression line. The significance level for the analysis was set at *p* = 0.05.

## 3. Results

Table 3 shows the YLDs for all conditions likely benefiting from rehabilitation. For instance, it shows that, from 1990 to 2017, a 66.2% growth was found in estimated YLD Counts likely benefiting from physical rehabilitation, with a significant and linear growth of more than 5.1 billion YLDs per year (99% CI: 4.8–5.4; *r*^2^ 0.99). While countries from all income levels had significant and linear growths, low-income countries more than doubled (111.5% growth) their YLD Counts, likely benefiting from physical rehabilitation for the 28-year period.

Table 3 also shows significant growths in estimated YLD Rates for the same time-period, worldwide and across countries of all income levels. However, for low-income countries, the growth was not linear (i.e., best fit in a logarithmic model (*r*^2^ = 0.5)) and was only significant at a 95% confidence level (*p* = 0.02). Of note, upper middle-income countries had a yearly growth of 42.6 YLDs per 100,000 people (99% CI: 38.1–47.2; *r*^2^ = 0.96), significantly higher than any of the comparators, i.e., no overlap among the 99% CIs.

On Age-standardized YLD Rates, which adjusted YLDs for population size and ageing, Table 3 shows that only high-income and lower middle-income countries had a significant change since 1990, but, while the latter had a linear growth (99% CI: 0.89–3.78; *r*^2^ = 0.46), the former had a logarithmic decrease (*r*^2^ = 0.86; *p* < 0.01). Overall, in this metric, we observed the smallest magnitude of changes: a maximum of 2.7% change from 1990 to 2017.

In a final metric, Table 3 shows that the Percentage of YLDs Benefiting from Physical Rehabilitation significantly and linearly increased across locations (all: *p* < 0.01; minimum *r*^2^ = 0.87). However, while high-income countries had the highest percent value in 2017 (48.6%), low and middle-income countries (LMICs) grew significantly on an order of up to seven times: 0.200 (99% CIs 0.185–0.215) versus 0.030 (99 CIs 0.024–0.037). Figure 1 provides a graphical representation of the data on this metric, while Appendix A for the three other YLD metrics.

Finally, Table 4 quantifies YLDs for a conservative subgroup of health conditions historically associated with physical rehabilitation need, i.e., key musculoskeletal and neurological conditions often resulting, primarily, in mobility impairments. The table reveals that a narrow set of conditions accounted for 225 billion YLDs in 2017, nearly two-thirds (65.6%) of the total physical rehabilitation needs (see Table 3). On the evolution over time, the subgroup shows a pattern of the results similar to that of the whole set of conditions, with a few differences relative to the larger group as follows.

First, YLD Rates for low-income countries grew over time in a linear shape, not logarithmic. Second, the percentage of YLDs Benefiting from Physical Rehabilitation did not significantly change for high-income countries for the subgroup of conditions, while it did in the main analysis. Lastly, in the subgroup of conditions, Age-standardized YLD Rates decreased significantly worlwide but increased significantly in low-income countries.

## 4. Discussion

This paper is, to our knowledge, the first systematic, secondary analysis of global physical rehabilitation needs, based on publicly-available data from the most comprehensive source of global epidemiological data to date, i.e., the GBD 2017. In addition to an over 66% global increase in YLD Counts from 1990 to 2017 (5.1 billion additional YLDs per year), we found a 17% increase in YLD Rates, i.e., per size of the population. Therefore, the growth of YLDs from all conditions likely benefiting from physical rehabilitation have outpaced that of the population. The key factor influencing per-capita growth has been the global population ageing, which is why the world’s Age-Standardized YLD Rates for all conditions potentially benefitting from physical rehabilitation did not significantly change over the studied time-period. Aging of the population is likely to continue to be a worldwide phenomena for the next several decades (e.g., global population with 65 years or more is predicted to double by 2050, while the overall population is expected to grow by less than 30% [82]). Global needs for physical rehabilitation are likely to continue to grow in tandem, with consequent increased demands for physical rehabilitation.

Growth in need for physical rehabilitation was particularly prominent in LMICs. For instance, in low-income countries, the YLD Counts more than doubled since 1990, likely due to population growth from high fertility rates as YLD Rates grew at a much less pronounced rate. In fact, increases in YLD Counts were inversely proportional to the countries’ income levels, such that the greatest increase in needs for physical rehabilitation was found in countries with lower income levels and the least rehabilitation infrastructure. Indeed, rehabilitation resources in many LMICs remain quite limited [7,19,26,83,84]. The WHO estimates that skilled rehabilitation professionals for many LMICs are currently about one-tenth of those required [7]. Hence, many argue that it is urgent to take action to supply LMICs with increased resources, [24,26,83,84,85,86,87], and this is especially so given the higher nominal increases of physical rehabilitation needs.

Nonetheless, we note that, in low-income countries, we also observed a logarithmic growth in YLD Rates, flattening around the year 2000 (see Appendix A for the graphical representation of the data, page 2). It is possible that global health activities for the Millennium Development Goals, which focused, for example, in preventable neonatal and infectious conditions, have helped control the rise in the prevalence of those conditions, and therefore the resulting YLDs in lower income countries [88]. This interpretation is reinforced by the data in Table 4. In that analysis, which focused on a narrow set musculoskeletal and neurological diseases and injuries (i.e., did not account for infectious or most neonatal conditions), we rather observed a linear instead of logarithmic growth in YLD Rates. Global development and advances in disease treatment and prevention in LMICs (e.g., targeting communicable, neonatal, and maternal conditions) result in higher life expectancy in these countries—hence higher likelihood of people experiencing, surviving, and living longer with chronic conditions or long-term disabilities, including those arising from age-related musculoskeletal or neurological conditions. [1,2,3] This epidemiological transition in LMICs has been observed for longer in high-income countries, and result in a trend for increasing YLD Rates, especially from musculoskeletal and neurogical conditions, which are not likely addressed by vertical, disease-focused approaches. Even infectious or neonatal conditions might not have been addressed in their sequalae (i.e., toward reducing or mitigating their disability through rehabilitation) as much as in its prevalence. For instance, while children with complex health conditions and people with HIV/AIDs increasingly survive in LMICs, the associated disabilities and rehabilitation needs remain often unmet [84,86,89,90,91,92]. In this scenario, granting access to needed rehabilitation (e.g., through horizontal, sustainable health systems’ strengthening [24]), especially in LMICs, is of utmost importance [24,85,87], and justified by the current YLDs data.

In addition to resource increases, innovative solutions might be further developed toward increasing access to timely and high quality rehabilitation services for locations with suboptimal or unevenly distributed rehabilitation resources. Potential solutions are needed for rehabilitation delivered at home and in the community, offering the potential that tele-rehabilitation may help fill the gap [43,86,93,94,95,96,97,98]. Task-shifting to healthcare assistants or team-based community care, especially in LMICs, is another potential option to meet global health needs in LMICs warranting future research and development [99].

Only in high-income countries did Age-standardized YLDs diminish over time, whereas in LMICs they did not significantly change or have significantly increased since 1990. This pattern was consistent for both of our analyses: that on the overall physical rehabilitation needs, and that focused on a narrow set of neurological and musculoskeletal conditions. We hypothesize that the wide implementation of public and private physical rehabilitation services in high-income countries, but to a far lesser extent in LMICs, has contributed to these findings. Historically, LMICs’ health systems and global health activities have been focused in reducing preventable mortality, while the global burden of non-fatal health losses has been lagging in terms of global health priority, resources, and gains [1,5,6,100]. Fortunately, there is now a global awareness about the need to avert preventable YLDs through appropriate healthcare services, including rehabilitation [5,6,24,101]. Similarly, global health policies are, now, more inclusive of people with disabilities and their rehabilitation needs [87,102,103]. Finally, the evidence of cost-effectiveness of rehabilitation is escalating [44,104]. In this context, our finding that Age-standardized YLDs was significantly reduced in high-income countries, but not in LMICs—and at times significantly increased in LMICs, can be yet another indicator of the need to scale up rehabilitation services in LMIC’s health systems [24]. Cardiac rehabilitation programs might be a good example, as they have been proven cost-effective [44], are generally standard practice in high-income countries, yet barely existent in many LMICs [105,106,107], even though service delivery models suitable to the LMICs have been developed [108].

Finally, we found the percentage of YLDs likely benefiting from physical rehabilitation increased significantly and linearly across analyzed locations. Furthermore, we observed a trend toward convergence, as the high-income countries had the highest value in 2017, but the growth in LMICs has been significantly higher since 1990. However, even in high-income countries, a significant growth has been persisting over time in the overall physical rehabilitation needs, indicating that the ceiling has not been reached yet in the percentage of YLDs Benefiting from Physical Rehabilitation. That ceiling, however, was found in the analysis narrowly focused on key musculoskeletal and neurological conditions. Therefore, in high-income countries, the areas of physical rehabilitation practice emerging in the more recent decades (e.g., focused on neoplasms or cardiothoracic conditions) have been, altogether, key for the continued growth in the percentage of YLDs Benefiting from Physical Rehabilitation. The higher the percent value, the higher the expected impact of physical rehabilitation, i.e., in terms of being able to reduce a higher share of the global burden of non-fatal health conditions.

### Limitations

The paper’s limitations are the following: First, the YLD measure is, at best, a proxy indicator of need for physical rehabilitation. For instance, functional data at the population level, e.g., from the Model Disability Survey, and/or other disability statistics, whenever available, might be used in alternative or complement to provide a more comprehensive picture of populations’ physical rehabilitation needs [85,109,110,111]. This limitation notwithstanding, the GBD is the most comprehensive source of global epidemiological data to date, and GBD data are widely used to analyze of the needs for, or benefits from, specific types of healthcare [112,113,114,115]. 

Second, there is no global standard of conditions that potentially benefit from physical rehabilitation which we could use for the selection of specific health conditions to include in our main analyses. To inform that selection, we relied on previous research on physical rehabilitation, a working definition of physical rehabilitation, and tailored searches on PubMed. Nonetheless, tailored searches conducted on PubMed, to identify and help select of health conditions likely benefiting from physical rehabilitation, do not equate to a systematic review itself. For conditions which were particularly uncertain with regards to potential benefits from physical rehabilitation, we employed explicit, a priori evidence-informed reasoning about the potential trade-offs for over- vs. under-estimation of physical rehabilitation needs.

Third, while a separate analysis for a subgroup of health conditions historically associated with physical rehabilitation need is present, it does not equate to minimum requirements in terms of a physical rehabilitation need.

Fourth, once we partly relied on published evidence to ground selection decisions and our trade-off reasoning, any under-research on the rehabilitation of conditions less common in high-income countries (e.g., tuberculosis, malaria) might have biased that reasoning. It is unknown whether the such putative under-research contributes under-estimate the magnitude of physical rehabilitation needs, especially in lower income countries.

Fifth, we used inferential statistics with YLD estimates, rather than direct data collection with metrics to quantify the nature and degree of physical disability. Moreover, the uncertainty intervals of the underlying YLD estimates could not be computed for our ‘combined’ values (i.e., the sum of YLDs from all health conditions likely benefeting from physical rehabilitation), although they exist in the public domain for each included condition in isolation [6].

Sixth, the quantity and accuracy of the data used to compute the GBD estimates is typically lower in lower income countries, which can include a less accurate detection or registry of data for many health conditions, especially non-communicable diseases or injuries. That can partly justify why the LMICs have much lower values for example in the Age-Standardized YLD Rates, across the time-series, when compared to high-income countries. Therefore, it is possible that the true magnitude of physical rehabilitation needs in LMICs remain partly obscure in these results. In addition, the quantity and accuracy of the underlying data in 2017 is likely superior to what it was back in 1990. This means a lower precision for the GBD estimates for the earlier years. However, lower preciseness does not imply systematic error (i.e., bias) toward over- or under-estimation of YLDs for the earlier times. To reduce potential bias, at each new cycle, the GBD study uses the more complete and updated data, measurement and classification systems, and estimation methods to re-calculate the entire time-series, not just the estimates for the more recent years [6].

Seventh, the presence of auto-correlation in the time series was not ruled out, i.e., we did not assess whether error terms were significantly correlated over time. Nonetheless, many of the simple linear regression models had nearly perfect fits with the underlying time series data, meaning that the magnitude of each of those errors was negligible. In addition, we used only simple linear regression analyses which did not account for what had changed over time apart from population growth and ageing. The latter ones here were inherently accounted for when we performed analyses using “Age-standardized YLD Rates” instead of simple YLD counts.

Finally, we focused on evolving trends for overall needs for physical rehabilitation and then in a narrow, conservative subgroup of conditions benefeting from physical rehabilitation, and did not perform sub-group analyses across specific groups of conditions (e.g., musculoskeletal, neurological, cardiac, etc.) as these have been performed by other groups that have used GBD data.

## 5. Conclusions

This systematic secondary analysis of the GBD 2017 provides global data on current physical rehabilitation needs, and on how those needs evolved since 1990, including across countries of varying income level.

Physical rehabilitation needs have been growing significantly over time across locations, not only in absolute terms, but also per-capita and in percentage of the total amount of total YLDs. This means that not only has physical rehabilitation been growing, but also that physical rehabilitation is now capable of averting a higher portion of the global burden of disability. The highest absolute growth of physical rehabilitation needs was observed in countries of lower income level, which typically have deprived rehabilitation infrastructure. Finally, only in high-income countries did a significant reduction in age-standardized needs per capita occur, which can be related to the wider implementation of physical rehabilitation services in these countries.

The use of the comprehensive, up-to-date GBD estimates for the needs-side of the rehabilitation resources-planning equation, albeit imperfect, is a readily available means to help health planners to meet the physical rehabilitation needs of the global population. These needs were found to be high and increasing, including per-capita, across countries of varying income level.

## Figures and Tables

**Figure 1 ijerph-16-00980-f001:**
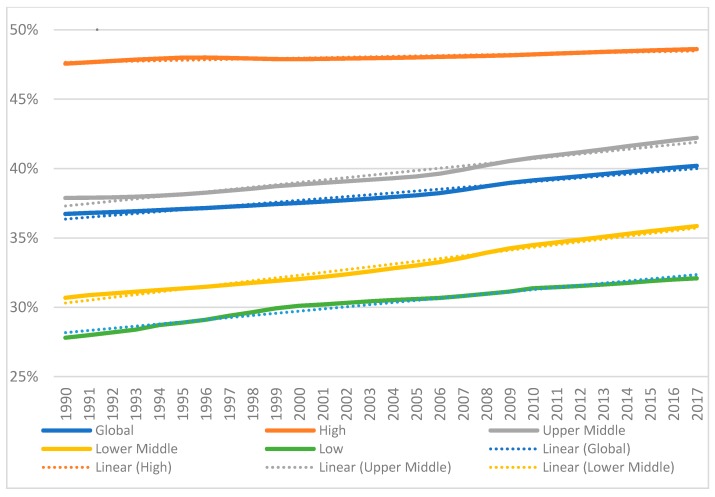
Percentage of YLDs Benefiting from Physical Rehabilitation among Total YLDs for the globe and across countries of varying income level. Linear regression lines (dotted) displayed.

**Table 1 ijerph-16-00980-t001:** List of GBD causes, injuries and impairments selected as generating YLDs that might potentially benefit from physical rehabilitation.

Causes
Communicable, Maternal, Neonatal or Nutritional:
• HIV/AIDs
• Leprosy; Zika
• Meningitis, Encephalitis; Tetanus
• Neonatal Disorders
Non-Communicable
• Neoplasms
• Cardiovascular Diseases (includes Stroke)
• Chronic Respiratory Diseases
• Neurological disorders, except Epilepsy and Migraine (tension-type headaches included)
• Autism Spectrum Disorder
• Musculoskeletal conditions (includes Low back Pain and Neck Pain)
• Congenital Birth Defects, except Urogenital and Digestive
**Injuries (nature of the)**
• Amputations
• Burns
• Fractures, except skull
• Head Injuries
• Spinal Injuries
• Minor Injuries: muscle and tendon injuries, including sprains and strains lesser dislocations; Open wound(s)
• Dislocation of hip; Dislocation of knee; and Dislocation of shoulder
• Asphyxiation
• Crush injury; Nerve Injury; Severe Chest Injury
• Multiple fractures, dislocations, crashes, wounds, pains, and strains
**Impairments (from the non-selected “causes” combined)**
• Heart Failure
• Guillain-Barré Syndrome

**Table 2 ijerph-16-00980-t002:** Subgroup of health conditions for a separate, highly conservative analysis of physical rehabilitation need.

Causes
Communicable, Maternal, Neonatal or Nutritional:
• Neonatal Disorders—only the Neonatal Encephalopathy due to Birth Asphyxia and Trauma
Non-communicable
• Stroke
• Neurological disorders, except Epilepsy, Dementia and Migraine
• Musculoskeletal conditions (includes Low back Pain and Neck Pain)
• Congenital Birth Defects—only Neuro Tube Defects + Congenital Musculoskeletal and Limb Anomalies
**Injuries (nature of the)**
• All of those in Table 1

**Table 3 ijerph-16-00980-t003:** Main analysis: YLDs from all conditions likely benefiting from Physical Rehabilitation (1990–2017).

Location/Item	# 1990	# 2017	% Change (1990–2017)	Regression Model Type	*r* ^2^	*b* Coefficient	95% CI	99% CI
YLD Counts, Billions								
**World**	206	343	66.2%	Linear	0.99	5.10 *	4.88–5.32	4.80–5.40
High-Income	58	79	37.4%	Linear	0.99	0.81 *	0.77–0.84	0.76–0.86
Upper Middle-Income	76	123	62.1%	Linear	0.99	1.78 *	1.69–1.87	1.66–1.90
Lower Middle-Income	62	119	90.4%	Linear	0.99	2.10 *	2.02–2.19	1.99-2.22
Low-Income	10	21	111.5%	Linear	1	0.39 *	0.38–0.40	0.38–0.41
**YLD Rates**								
**World**	3825	4488	17.3%	Linear	0.96	25.7 *	23.7–27.7	23.0–28.4
High-Income	5748	6643	15.6%	Linear	0.98	33.1 *	31.1–35.1	30.4–35.8
Upper Middle-Income	3594	4669	29.9%	Linear	0.96	42.6 *	39.3–46.0	38.1–47.2
Lower Middle-Income	3233	3806	17.7%	Linear	0.96	21.6 *	19.9–23.2	19.3–23.8
Low-Income	2977	3112	4.5%	Logarithmic	0.50	2.5 **	0.40–4.55	−0.33–5.28
**Age-Standardized YLD Rates**								
**World**	4377	4334	−1.0%	Logarithmic	0.22	−0.62	−2.13–0.89	−2.66–1.42
High-Income	5007	4872	−2.7%	Logarithmic	0.86	−5.36 *	−6.76–(−3.96)	−7.26–(−3.47)
Upper Middle-Income	4106	4080	−0.6%	Linear	0.04	1.34	−1.38–4.06	−2.33–5.02
Lower Middle-Income	4262	4314	1.2%	Linear	0.46	2.33 *	1.26–3.40	0.89–3.78
Low-Income	4189	4276	2.1%	Logarithmic	0.15	0.29	−3.29–3.87	−4.55–5.14
**% of YLDs Benefiting from Physical Rehabilitation (among Total YLDs)**								
**World**	36.7%	40.2%	9.5%	Linear	0.97	0.135 *	0.125–0.145	0.122–0.148
High-Income	47.6%	48.6%	2.2%	Linear	0.87	0.030 *	0.026–0.035	0.024–0.037
Upper Middle-Income	37.9%	42.2%	11.4%	Linear	0.97	0.170 *	0.157–0.183	0.153–0.187
Lower Middle-Income	30.7%	35.9%	16.8%	Linear	0.98	0.200 *	0.189–0.211	0.185–0.215
Low-Income	27.8%	32.1%	15.4%	Linear	0.97	0.155 *	0.145–0.165	0.141–0.169

Data obtained from: http://ghdx.healthdata.org/gbd-results-tool. Abbreviations: YLD—Year Lived with Disability. Legend: * *p* < 0.01, ** *p* < 0.05. Notes: The “*b* coefficient” refers to the annual change within the linear regression model. Different population structures apply to countries with varying income levels, so cross-location comparisons are not valid for the metric YLD Counts.

**Table 4 ijerph-16-00980-t004:** YLDs from a narrow set (i.e., subgroup) of health conditions benefiting from Physical Rehabilitation (1990–2017).

Location/Item	# 1990	# 2017	% Change (1990–2017)	Regression Type: Best Fit	*r* ^2^	*b* Coefficient	95% CI	99% CI
YLD Counts, Billions								
**World**	135	225	66.6%	Linear	0.99	3.32 *	3.18–3.43	3.13–3.51
High-Income	40.3	54.7	35.6%	Linear	0.99	0.52 *	0.50–0.55	0.49–0.56
Upper Middle-Income	49.7	82.8	66.5%	Linear	0.99	1.24 *	1.19–1.29	1.17–1.31
Lower Middle-Income	38.8	73.9	90.6%	Linear	0.99	1.30 *	1.24–1.36	1.22–1.38
Low-Income	5.9	12.5	113.6%	Linear	0.98	0.24 *	0.23–0.25	0.22–0.26
**YLD Rates**								
**World**	2506	2942	17.4%	Linear	0.96	16.6 *	15.3–17.8	14.8–18.3
High-Income	4030	4597	14.1%	Linear	0.97	19.7 *	18.3–21.1	17.8–21.6
Upper Middle-Income	2356	3143	33.4%	Linear	0.98	30.8 *	28.9–32.6	28.3–33.2
Lower Middle-Income	2009	2367	17.8%	Linear	0.98	13.2 *	12.0–14.4	11.6–14.9
Low-Income	1776	1875	5.6%	Linear	0.68	3.08 *	2.22–3.94	1.92–4.24
**Age-Standardized YLD Rates**								
**World**	2863	2812	−1.8%	Logarithmic	0.61	−1.64 *	−2.66–(−0.61)	−3.02–(−0.25)
High-Income	3504	3397	−3.1%	Logarithmic	0.83	−5.25 *	−6.48–(−4.02)	−6.92–(−3.59)
Upper Middle-Income	2664	2677	0.5%	Linear	0.18	1.72 **	0.23–3.22	−0.30–3.75
Lower Middle-Income	2657	2665	0.3%	Linear	0.03	0.38	−0.44–1.20	−0.72–1.48
Low-Income	2564	2656	3.6%	Linear	0.82	3.33 *	2.70–3.95	2.48–4.17
**% of YLDs Benefiting from Physical Rehabilitation (among Total YLDs)**								
**World**	24.1%	26.4%	9.5%	Linear	0.97	0.085 *	0.079–0.092	0.077–0.094
High-Income	33.3%	33.6%	0.9%	Linear	0.12	0.005	−0.011–0.0005	−0.013–0.003
Upper Middle-Income	24.8%	28.4%	14.4%	Linear	0.99	0.137 *	0.131–0.143	0.129–0.145
Lower Middle-Income	19.1%	22.3%	16.8%	Linear	0.97	0.122 *	0.113–0.131	0.110–0.134
Low-Income	16.6%	19.4%	16.5%	Linear	0.95	0.107 *	0.097–0.117	0.093–0.120

Data obtained from: http://ghdx.healthdata.org/gbd-results-tool. Abbreviations: YLD—Year Lived with Disability. Legend: * *p* < 0.01, ** *p* < 0.05. Notes: The “*b* coefficient” refers to the annual change within the linear regression model. Different population structures apply to countries with varying income levels, so cross-location comparisons are not valid for the metric YLD Counts.

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
