# Peer review of "Global Need for Physical Rehabilitation: Systematic Analysis from the Global Burden of Disease Study 2017"

_ijerph, 2019, doi:10.3390/ijerph16060980_

Reviewer 1 Report

Thank you for the opportunity to review this article.

My comments:

First, while you cover it extensively after Table 1, I remain unconvinced that inclusion of what you call “potentially dubious” causes that are “amenable to physical rehabilitation” is warranted. Since it is not possible to determine the amount or proportion of YLDs from each cause that are “amenable to physical rehabilitation,” I would encourage you to be conservative and only include those items (especially injuries, musculoskeletal conditions, etc…) that are more clearly connected to physical rehab. I kept thinking to myself, “what % of cardiovascular YLDs should be included?” Since this cannot be answered I’d suggest using more conservative criteria. At a minimum, run more conservative models and see if there are major differences.

In the data extraction section, the second paragraph describes background on how YLDs are calculated, not how you extracted the data. In this section please provide a better description of what your dataset looks like. Do you have countries, years, income classification, and then the 4 YLD measurements? (or do you ignore countries? If so, why?) Then, in the data analysis section explain your model better. What’s the unit of analysis, sample size, and what exactly are you testing for significance? (That there was a non-zero annual increase.) R-squared means something very different here than it does in multivariable analysis, be sure not to overstate its meaning.

Please include at least 1 graph of the 4 different YLD measurements (perhaps on the same figure) over time in the text, not just in the appendix.

Something to consider in limitations:

How much of the increase in YLDs could be attributed to better measurement/recognition of different causes/injuries/impairments over time?

Would autocorrelation be an issue in this time series analysis?

Author Response

Our comments follow in red, while those from the reviewers in black. We’d like to thank both reviewers for their reviews and suggestions. They resulted into substantive additions and improvements in the clarity of manuscript.  Their feedback clearly helped to strengthen the manuscript. All of the resulting changes were tracked for easy detection, except for the inclusion or redesign of new tables – in which tracking the new context would be visually confusing. Indeed, we now add one new table in the Methods and another in the Results, as well as have completed redesigned the structure of a previous table – the one containing the main results. We below answer to each of the reviewers’ comments. The references to line numbers report to the tracked version of the resubmitted manuscript.

REVIEWER 1

Thank you for the opportunity to review this article.

My comments:

First, while you cover it extensively after Table 1, I remain unconvinced that inclusion of what you call “potentially dubious” causes that are “amenable to physical rehabilitation” is warranted. Since it is not possible to determine the amount or proportion of YLDs from each cause that are “amenable to physical rehabilitation,” I would encourage you to be conservative and only include those items (especially injuries, musculoskeletal conditions, etc…) that are more clearly connected to physical rehab. I kept thinking to myself, “what % of cardiovascular YLDs should be included?” Since this cannot be answered I’d suggest using more conservative criteria. At a minimum, run more conservative models and see if there are major differences.

We appreciate your comment. We felt puzzled in a similar way, and as mentioned there is no global standard we could rely on. Hence, we decided to follow a process that did not under-estimate physical rehabilitation needs on one hand, and not over-estimate in the other, relative to the state of knowledge on the effectiveness of physical rehabilitation. In that context, our option was to actively search for the current aggregated evidence (e.g. preferably Cochrane reviews) on the benefit (hopefully cost-effectiveness) of physical rehabilitation on any “dubious” health conditions – and we now define what that means (Methods: lines 113-114). From that process, some conditions were finally included, others not. The Appendix 1 provides our evidence-informed trade-off reasoning relative to all the “potentially dubious” inclusions. Albeit imperfect, we kept those premises for our main analysis.

That being said, we find value in the suggestion of the reviewer. As such, we now additionally present a way more conservative analysis, using a subgroup of health conditions - focused on key musculoskeletal and neurological diseases and injuries. As a result of that additional analysis, substantive additions are now included in the Methods (Lines 141-145), Results (Lines 223-233) including the whole new Table 4 (Lines 238-243), and in the Discussion (Lines 273-275, 289-291 and 311-316), including in the Limitations (Lines 335-337). Overall, the suggestion clearly strengthened the paper.

In the data extraction section, the second paragraph describes background on how YLDs are calculated, not how you extracted the data. In this section please provide a better description of what your dataset looks like. Do you have countries, years, income classification, and then the 4 YLD measurements? (or do you ignore countries? If so, why?) Then, in the data analysis section explain your model better. What’s the unit of analysis, sample size, and what exactly are you testing for significance? (That there was a non-zero annual increase.) R-squared means something very different here than it does in multivariable analysis, be sure not to overstate its meaning.

In this revised version, we produced substantial changes in the reporting of the Methods section, notably between lines 148 to 192. This came as the combined result of the valuable comment from both reviewers. We hope the changes made address the issues raised. Specifically regarding the R-Squared issue, we used it essentially as an indicator (and not the only one) for determining the fit of the regression model. In the new structure (i.e. table design) to report the result, we provide more emphasis in the b-coefficient and the respective confidence intervals, at both the 95% and 99% confidence level.

Please include at least 1 graph of the 4 different YLD measurements (perhaps on the same figure) over time in the text, not just in the appendix.

We appreciate this valuable suggestion, and in the revised version we present one of the graphics in the main text, and also outline now that those for the remaining metrics are available in an Appendix. Before, we did not make such a call in the Results. Changes were reflected in lines 221-222 and lines 234-236.

Something to consider in limitations:

How much of the increase in YLDs could be attributed to better measurement/recognition of different causes/injuries/impairments over time?

Would autocorrelation be an issue in this time series analysis?

These are quite pertinent issues. We now address both in the Limitations section, specifically from lines 347 to lines 365.

Reviewer 2 Report

Thank you for the opportunity to review “ Global need for physical rehabilitation: systematic analysis from the Global Burden of Disease Study 2017” for Int. J. Environ. Res. Public Health. This paper analyzes evolving trends in physical rehabilitation needs, using data on Years Lived with Disability (YLDs) from the Global Burden of Disease Study (GBD) 2017. Overall, the paper is very well-written. But there are some comments below for the authors to consider:

Introduction

1. I thought the first research questions would sound better if it started as “How large instead of How many,” but this optional of course.

Methods

2. I would recommend either change the heading “Data selection” into “Measures” or create a new heading and provide definition and how each outcome was measured.  

a. Amenable to Physical Rehabilitation YLDs.

b. YLDs amenable to Physical Rehabilitation rate per 100 000 people

c. Age-standardized Amenable to Physical Rehabilitation YLDs rate per 100 000 people

d. Percentage of amenable to Physical Rehabilitation YLDs for the respective condition among the total number of YLDs

Data extraction text can be used for the Measures section.

Data analysis

1. I think what you tried to do is the measure the changes in outcomes between 1990 and 2017. If two measures used then you could just report t-test was used to testing the significance or ANOVA if all repeated measures were used. I would also recommend creating two tables instead of Table 2. Table 1 as descriptive statistics for changes in trends between 1990 and 2017. Hence, you need to describe outcome values in 1990. For the column of change, you can add * if p<0.01 and ** if p<0.05.< p="">

2. Next, you just calculated the average annual growth in your outcomes during the 28-year time period using linear regression analysis (OLS). I think it would be best to use only OLS instead of also using a logged form of OLS for a couple of your outcomes. This helps more straightforward interpretation of the results (average annual change ). You can use the next table to report the annual average changes per year based on OLS results. You should have a column for OLS B coefficient and the 95%CI separately. You can also add starts to b-coefficient if they are significant 

3. I think saying” We tested for two hierarchical levels of statistical significance: 99 and 95%, respectively” is rather wordy and it would be sufficing to say that the significance level for the analysis was set at p=0.05. In the table then you show that *means p<0.01 and ** means p<0.05.< p="">

Results

1. If these changes are acceptable, hence authors need to report the 28-year period changes and indicate outcome values for both 1990 vs. 2017, as per the new table discussed above.

I thought, authors, to also discuss the population structure between high-income countries 

For all Tables

1. All tables should have a clear heading on the top of the table. Add the source of data at the bottom of the table. 

2. Add any abbreviations/acronyms used in the text, e.g., YLD, etc. 

3. Add any further footnote to describe the data, e.g. *p<0.01, **p<0.05, etc. Ideally, each table should be stand-alone information.

Author Response

Our comments follow in red, while those from the reviewers in black. We’d like to thank both reviewers for their reviews and suggestions. They resulted into substantive additions and improvements in the clarity of manuscript.  Their feedback clearly helped to strengthen the manuscript. All of the resulting changes were tracked for easy detection, except for the inclusion or redesign of new tables – in which tracking the new context would be visually confusing. Indeed, we now add one new table in the Methods and another in the Results, as well as have completed redesigned the structure of a previous table – the one containing the main results. We below answer to each of the reviewers’ comments. The references to line numbers report to the tracked version of the resubmitted manuscript.

REVIEWER 2

Comments and Suggestions for Authors:

Thank you for the opportunity to review “ Global need for physical rehabilitation: systematic analysis from the Global Burden of Disease Study 2017” for Int. J. Environ. Res. Public Health. This paper analyzes evolving trends in physical rehabilitation needs, using data on Years Lived with Disability (YLDs) from the Global Burden of Disease Study (GBD) 2017.Overall, the paper is very well-written.

Thank you for your comment.

But there are some comments below for the authors to consider:

Introduction
1. I thought the first research questions would sound better if it started as “How large instead of How many,” but this optional of course.
We follow the reviewer’s suggestion and have changed accordingly – see in line 74. 

Methods

2. I would recommend either change the heading “Data selection” into “Measures” or create a new heading and provide definition and how each outcome was measured.  

a. Amenable to Physical Rehabilitation YLDs.

b. YLDs amenable to Physical Rehabilitation rate per 100 000 people

c. Age-standardized Amenable to Physical Rehabilitation YLDs rate per 100 000 people

d. Percentage of amenable to Physical Rehabilitation YLDs for the respective condition among the total number of YLDs

Data extraction text can be used for the Measures section.

Along with other comments from reviewer 1, we have used these suggestions, nearly entirely, to improve the clarity of the reporting in the Methods section. We believe it is strengthened now – see from lines 148 to 174.

Data analysis

1. I think what you tried to do is the measure the changes in outcomes between 1990 and 2017. If two measures used then you could just report t-test was used to testing the significance or ANOVA if all repeated measures were used. I would also recommend creating two tables instead of Table 2. Table 1 as descriptive statistics for changes in trends between 1990 and 2017. Hence, you need to describe outcome values in 1990. For the column of change, you can add * if p<0.01 and ** if p<0.05.< p="">

2. Next, you just calculated the average annual growth in your outcomes during the 28-year time period using linear regression analysis (OLS). I think it would be best to use only OLS instead of also using a logged form of OLS for a couple of your outcomes. This helps more straightforward interpretation of the results (average annual change ). You can use the next table to report the annual average changes per year based on OLS results. You should have a column for OLS B coefficient and the 95%CI separately. You can also add starts to b-coefficient if they are significant 

3. I think saying” We tested for two hierarchical levels of statistical significance: 99 and 95%, respectively” is rather wordy and it would be sufficing to say that the significance level for the analysis was set at p=0.05. In the table then you show that *means p<0.01 and ** means p<0.05.< p="">

Once again, these suggestions were highly instrumental for us – notably to help improve the reporting of the procedures for the data analysis (from lines 179 to 192) and then the reporting of the respective results (from lines 211 to 216). We do not follow specifically the suggestion to split the previous results into two tables, since we now have another table with supplementary results (as a result of the comments of the 1st reviewer). Overall, we wanted to avoid a further proliferation in the number of tables. Nonetheless, we did re-design the table of the results, for including and/or separating the information overall in the way this reviewer has suggested. We believe the table is now more informative, has a cleaner look, and is altogether much easier to read and interpret.

4. I think saying” We tested for two hierarchical levels of statistical significance: 99 and 95%, respectively” is rather wordy and it would be sufficing to say that the significance level for the analysis was set at p=0.05. In the table then you show that *means p<0.01 and ** means p<0.05.< span="">

This was changed accordingly – lines 191-192.

Results

1. If these changes are acceptable, hence authors need to report the 28-year period changes and indicate outcome values for both 1990 vs. 2017, as per the new table discussed above.

I thought, authors, to also discuss the population structure between high-income countries 

For all Tables

1. All tables should have a clear heading on the top of the table. Add the source of data at the bottom of the table. 

2. Add any abbreviations/acronyms used in the text, e.g., YLD, etc. 

3.  Add any further footnote to describe the data, e.g. *p<0.01, **p<0.05, etc. Ideally, each table should be stand-alone information.

We now address each of these numbered issues for our Results, specifically in the tables. Thank you.

Round  2

Reviewer 1 Report

It’s much better this time around. I appreciate the changes to the methods section in particular but still have a few recommendations.

In the new section (lines 148-174) no need to use “_____” for words like age, metrics, etc… Some typos in this section as well.

Lines 154-157, overused “sequalae”

Lines 163-165 These are YLDs just for diseases benefitting from physical rehab, right?

In the results section lines 166-168, as opposed to the other 3 sets of indicators, you calculate the % of YLDs benefitting from phys. therapy. For this indicator, I presume you also used the total number of YLDs which you extracted for your denominator. Be precise and comprehensive in your description of the process.

You could probably combine tables 3 and 4 (and also combine tables 1 and 2). There isn’t as much of a need to talk about the regression results, CI, r^2, as you’re really looking at trends over time. I think you make your main points well enough with just the % changes and, if you insist, the beta coefficients and significance at the .05 level.

Lines 229-233 The differences between your robustness check and the initial analysis are minimal. The choice between linear and logarithmic models is not helpful as a difference. Growth patterns are very similar and this would be easier to see this if you combined the tables.

In the figures, the dotted lines for the regression fit aren’t really helpful. They don’t tell us anything we can’t already see from the actual trend line. I’d remove them. Also, choose different colors. The two blue colors are too similar. The additional “High-income: Growth per year…” inside of the figure is distracting and we already have this information in the table. I’d take those out as the figures are too busy.

Lines 268-9 I’m still not convinced that the logarithmic growth vs. nearly linear growth argument is strong. The lines look pretty straight except in figure 3. Again, I don’t think the log vs. lin argument really matters for your overall case, and would consider dropping it entirely.

If people are living longer with disabilities, especially in settings where it was difficult to live longer in the past few decades, how does that affect your measurement vs. what’s happening in more developed settings where it may have already been feasible to live longer with disabilities?

Author Response

Our response follows in red. We would like to thank the reviewer for the further suggestions; many of which resulted in substantive changes and improvements in the paper. While we track the written changes we make in the paper as a result of the comments, we do not track (yet mention below) any changes we have made in the tables or figures.

Comments and Suggestions for Authors

It’s much better this time around. I appreciate the changes to the methods section in particular but still have a few recommendations.

Thank you for your comment. Much appreciated.

In the new section (lines 148-174) no need to use “_____” for words like age, metrics, etc… Some typos in this section as well.

Thank you for noting these issues. We proceeded accordingly. Changes from line 149 to 172.

Lines 154-157, overused “sequalae”.

We now replace the term a number of times in the newly revised version to avoid overuse. Changes within the line 157.

Lines 163-165 These are YLDs just for diseases benefitting from physical rehab, right?

Yes. We did not specify it before, but we do so now in this revised version. It is important, and we thank you.

In the results section lines 166-168, as opposed to the other 3 sets of indicators, you calculate the % of YLDs benefitting from phys. therapy. For this indicator, I presume you also used the total number of YLDs which you extracted for your denominator. Be precise and comprehensive in your description of the process.

We did not need to extract denominators and we did not calculate percentages; the metric percentage – for which we extracted data – is performed automatically. However, we did fail to report before the exact meanings of the metrics for which we have extracted data, including that percentage metric. We close that gap now and have been much more explicit: lines 161-163.

You could probably combine tables 3 and 4 (and also combine tables 1 and 2). There isn’t as much of a need to talk about the regression results, CI, r^2, as you’re really looking at trends over time. I think you make your main points well enough with just the % changes and, if you insist, the beta coefficients and significance at the .05 level.

We respectfully would like to include both Tables 3 and 4.  While we agree that % changes are illustrative in many occasions, that is not the case for all instances. For example, in table 3 (our main analysis), we observe the following:

#

1990

#

2017

%   change

[1990-2017]

Regression   Model Type

b  

coefficient  

95%   CI

99%   CI

Age-standardized YLD Rates

Lower Middle-Income

4262

4314

1.2%

Linear

.46

 2.33 *

1.26–3.40

0.89–3.78

Low-Income

4189

4276

2.1%

Logarithmic

.15

0.29

-3.29–3.87

-4.55–5.14

If we just rely in the % change value, we would understand that low-income countries had nearly twice the % change in their Age-standardized YLD Rates from 1990 to 2017 relative to lower middle-income countries. However, if we observe all the other elements (regression model type with the best fit, r2, b coefficients and CIs) we get a much more comprehensive view of the evolving trend over time, i.e. of what happened in all years between 1990 and 2017, for which we extracted, processed and analyzed the data: see Methods, lines 186-189. Therefore, with the full elements analyzed, we have a different reading. We observe that the lower middle-income countries had a statically significant yearly growth (at 99% confidence level) in a relatively linear way, completely at the odds of low-income countries.

Indeed, beyond only 2 data points (1990 and 2017), the pattern between all the 28 data points (i.e. the repeat observations for each year between 1990 and 2017) is of statistical relevance and is better appraised, we believe, by the full elements of the table, as well through the figures we also provide. With such regards, the reviewer later rightly notes that we repeat, in the figures, many of the results in the table. That is completely true. So, in that whole context, we preferred to delete the duplicative content rather in the figures, as some only follow in the Appendix, and preferred to retain the full content in the tables, which are part of our statistical analyses and main results.  We appreciate the feedback provided in this case.

Finally, we understand that the suggestion to combine tables 3 and 4 follow the assumption that a few elements of both tables would be deleted. As our preference is to retain all of the data in the tables (and eliminate the duplicative content in the figures), merging table 3 and 4 would be confusing (too many numbers and items).  For congruence, if we keep table 3 and 4 separated (also for conceptual reasons as we detail below), we believe that table 1 and table 2 benefit from remaining separate too, as table 1 provides the conditions for the results in the table 3 and table 2 provides the conditions for the results in table 4. Otherwise, the reader may become confused of which results pertain to each set of conditions and exactly where the separation of both occurs.

Lines 229-233 The differences between your robustness check and the initial analysis are minimal. The choice between linear and logarithmic models is not helpful as a difference. Growth patterns are very similar and this would be easier to see this if you combined the tables.

It is true that we observe and report that the differences are minimal (lines 230-236). We agree that those analyses indeed provide robustness to our results, especially in relation to our initial submission.

Given we provide two tables in the exact same form and structure, we believe it facilitates any direct comparisons.

Perhaps more importantly, we’d like to keep both analyses and tables separate for conceptual reasons. Table 3 is our main analysis (and is so identified in the header), while table 4 are supplementary analyses both supporting the robustness of our main findings and likely of interest to clinicians whose work is focused on those conditions. However, the supplementary analyses do not equate to physical rehabilitation needs, even ‘minimum requirements’ of physical rehabilitation needs as we outline in the Limitations (lines 349-351). As a means to emphasize that important point, we would prefer to keep the main analysis and the supplementary analyses in two separated tables, although we appreciate the suggestion.

 Re: Logarithmic vs linear models - see response below.

Lines 268-9 I’m still not convinced that the logarithmic growth vs. nearly linear growth argument is strong. The lines look pretty straight except in figure 3. Again, I don’t think the log vs. lin argument really matters for your overall case, and would consider dropping it entirely.

Indeed, the distinction between logarithmic growth vs. nearly linear growth seems stronger in figure 3. However, even in figure 2, there is a logarithmic (clearly not linear) type of growth for low-income countries between 1990 and 2017. It is, however, a bit hard to see that in the previous figure, as the scale is of the Y axis is necessarily large. Yet, when we zoom in over the YLD Rate values of low-income countries, we observe that logarithmic type of growth much better. Therefore, we do so zoom in the Appendix (page 2) to better provide the visual depictions.

Indeed, as we can there observe, the values around the year 2000 are even higher than in 2017, after a fast growth since 1990. The trend between 1990 and 2017 was not for a linear growth. It is worth-noting, however, that in the analysis focused only on the set of conditions historically associated to physical rehabilitation need, the low-income countries rather had a linear type of growth in that metric, as we report in the Results (line 232). That difference was an important matter for Discussion, which is now reinforced (272-293).

Those important discussion points would have been dropped altogether if the distinction between the different types of growth were not made and observed.

In the figures, the dotted lines for the regression fit aren’t really helpful. They don’t tell us anything we can’t already see from the actual trend line. I’d remove them. Also, choose different colors. The two blue colors are too similar. The additional “High-income: Growth per year…” inside of the figure is distracting and we already have this information in the table. I’d take those out as the figures are too busy.

Thank you for these excellent suggestions. We eliminated the additional information that is repeated from the tables. Also, we have changed the color set for better differentiation, i.e. changed the light blue for a green color. However, we retained the dotted lines, which visually represents the regression lines. While at times they almost completely overlap with the actual trends, at other times differences are more visible, especially in the Supplementary Figure 3 in the Appendix. Overall, we believe that the dotted lines help visualize the fit between the actual trend and the regression model. That is now, we believe, less visually confusing as other elements of the figures were eliminated as suggested.

If people are living longer with disabilities, especially in settings where it was difficult to live longer in the past few decades, how does that affect your measurement vs. what’s happening in more developed settings where it may have already been feasible to live longer with disabilities?

This is an interesting point. LMICs seem to increasingly experience the same kind of epidemiological transition that high-income countries experienced for longer time. So, increasingly the focus is for countries’ health systems to assure that the gain of years of life, inclusively of those who experience a disability, is complemented with the services people need to live well, not merely longer. We added content on the Discussion which help emphasize that – see line 279-293.